# Easy and Fast Production of Solketal from Glycerol Acetalization via Heteropolyacids

**DOI:** 10.3390/molecules27196573

**Published:** 2022-10-04

**Authors:** Diana Julião, Fatima Mirante, Salete S. Balula

**Affiliations:** LAQV/REQUIMTE & Department of Chemistry and Biochemistry, Faculty of Sciences, University of Porto, 4169-007 Porto, Portugal

**Keywords:** glycerol, acetalization, solketal, heteropoly acids

## Abstract

This work presents an effective and fast procedure to valorize the main waste produced from the biodiesel industry, i.e., the glycerol. The acetalization of glycerol with acetone represents an effective strategy to produce the valuable solketal, a fuel additive component. In this work, the catalytic efficiency of different commercial heteropolyacids (HPAas) was compared under a solvent-free system. The HPAs used were H_3_[PW_12_O_40_] (PW_12_), H_3_[PMo_12_O_40_] (PMo_12_) and H_4_[SiW_12_O_40_] (SiW_12_). The influence of reactional parameters such as reactants stoichiometry, catalyst concentration and reaction temperature were investigated in order to optimize experimental conditions to increase cost-efficiency and sustainability. HPAs demonstrated to be highly efficient for this type of reaction, presenting a high and fast glycerol conversion, with high selectivity to solketal under sustainable conditions (solvent-free system and room temperature medium). The activity of HPAs using 3% to glycerol weight and a glycerol/acetone ratio of 1:15 followed the order: PW_12_ (99.2%) > PMo_12_ (91.4%) > SiW_12_ (90.7%) as a result of the strong acidic sites after 5 min. In fact, only 5 min of reaction were needed to achieve 97% of solketal product in the presence of the PW_12_ as a catalyst. This last system presents an effective, selective and sustainable catalytic system to valorize glycerol.

## 1. Introduction

Over the past 200 years, human activity resulting from the constant growth and expansion of technology has been extremely dependent on petroleum exploitation [1]. The depletion of fossil fuels resources and the related environmental problems have encouraged the search for alternative energy sources. Biodiesel, which is a renewable fuel produced from the transesterification of vegetable oils or animal fats, has been demonstrating to be a valuable alternative to fossil fuels, with several environmental benefits [2,3]. The high promotion of biofuels by EU through energy-climate policies, fiscal incentives and supporting research projects, have remarkably encouraged the biodiesel production and, consequently, a serious surplus of glycerol (main by-product). The purification of crude glycerol to a chemically pure substance results in a valuable industrial chemical. However, purification is costly and the glycerol market is already saturated. Many studies have reported the glycerol as a promising raw material for producing value-added chemicals for a wide range of applications by various catalytic pathways involving oxidation, esterification, acetalization, etherification, hydrogenolysis and reforming [3,4,5,6,7,8,9,10,11]. Acetals and ketals are promising chemicals that can be obtained by the acetalization of glycerol with ketones [3,7,12]. Acetals are high-value fuel additives, increasing the octane number of the gasolines, and when added to diesel, biodiesel or mixtures, significantly reduce particulate and oxides emissions [3,4]. One promising additive is the 2,2-dimethyl-1,3-dioxolan-4-methanol, known as solketal, whose preparation involves the need for catalytic acidic sites [3,13,14]. Here, heterepolyacids (HPAs) with a Keggin structure can have a proeminent role due to their strong acidic sites and tuneable acidity, substituting the traditional environmental harmful and corrosive acids (e.g., HCl, HF, H_2_SO_4_, H_3_PO_4_, FeCl_3_ and SnF_2_) [2,13,15]. Moreover, their strong structure, inherent to the Keggin general formula [XM_12_O_40_]^n−^ (X = heteroatom, M = addenda atom), is an added value, avoiding a possible drawback resulting from the production of water [3]. The application of HPAs as catalysts in a glycerol acetalization reaction are rarely reported in the literature. Da Silva et al. reported the activity of various Brønsted acid catalysts as HPAs on glycerol ketalization with different ketones at room temperature under a solvent-free system [16]. In this work, the H_3_PW_12_O_40_ catalyst was much more active than other Brønsted acid catalysts (85% conversion achieved after 2 h and 97% of selectivity to solketal). Further, the catalyst was recovered and reused without loss of activity [16]. Some years later, the same authors demonstrated that the Sn_2_SiW_12_O_40_ compound is more active (> 99% conversion, after 1 h reaction at room temperature and 97% selectivity to solketal) than the corresponding acid H_4_SiW_12_O_40_ and other heteropoly catalysts [13]. Some other studies used the cesium salt of the polyoxotungstate as a catalyst for the acetalization of glycerol without the use of an additional solvent. A high efficiency was achieved after 1 h at room temperature [2] and the reaction time was decreased to 15 min when the homogeneous polyoxotungstate catalyst was immobilized in a Mesoporous silica [2].

The main goal of this work is to perform a careful comparative work between three different heteropolyacids (phosphotungstic acid (H_3_PW_12_O_40_), silicotungstic acid (H_4_SiW_12_O_40_) and phosphomolybdic acid (H_3_PMo_12_O_40_) to select the most active and selective for the acetalization of glycerol without the need of using an additional solvent. Further, the influence of reaction parameters, such as catalyst amount, molar ratio of the reactants and reaction temperature were studied to enhance conversion and selectivity to form solketal under a sustainable and cost-effective catalytic system.

## 2. Results and Discussion

The commercial heteropolyanions (HPAs) H_3_[PW_12_O_40_], H_3_[PMo_12_O_40_] and H_4_[SiW_12_O_40_], denoted as PW_12_, PMo_12_ and SiW_12_, respectively, were tested as catalysts for the acetalization of glycerol with acetone. The industrial production of fuel additives is associated to economic and environmental barriers. Therefore, it is important for the improvement of glycerol conversion with high solketal yield and short reaction times, by using effective catalysts with solvent-free conditions and mild temperatures. The effect of reaction conditions for glycerol acetalization with acetone was first explored using the PW_12_ as the acid catalyst. This reaction provides two cyclic products (Figure 1), i.e., the five-membered solketal (1) and the six-membered acetal (2).

The effect of the reaction temperature, evaluated in terms of glycerol conversion and selectivity was studied using two different temperatures: 25 °C and 60 °C. Others operating conditions were maintained: glycerol/acetone = 1:6 and 0.057 mmol of PW_12_. First, the reactional mixture was left under stirring during 10 min, and then the catalyst was added, starting the reaction (Figure 2a). The blank reactions were performed under the same conditions but in the absence of PW_12_. In the absence of the catalyst, the mixture of the glycerol/acetone was revealed to be immiscible at 25 °C, in which was observed a clear separation of two phases (glycerol and acetone) preventing the accurate study of the blank reaction. This behavior was not observed when in the presence of a catalyst, probably to an increase in acetone polarity resulting from the dissolution of PW_12_, which favored the miscibility between the two phases [16]. Figure 2 presents the glycerol conversion and solketal (1) selectivity obtained under each reaction temperature and during the blank study after 5 min of reaction. The acetalization reaction revealed to depend on the temperature (Figure 2b), obtaining a high glycerol conversion and consequently a superior propensity to obtaining solketal (1) as majority product at room temperature (81.7%) than at 60 °C (73.8%). Moreover, the formation of acetal (2) was more notorious in the blank reaction, demonstrating that the formation of five-membered rings (1); however, still formed in small amounts under a catalytic glycerol acetalization system (Figure 2a). Identical behavior for the products identification was reported previously in the literature [17,18]. 

The acetalization reaction is a reversible process; using an excess of acetone, with respect to glycerol, it will shift the equilibrium to the formation of products [19]. To investigate this parameter, the effect of the glycerol/acetone molar ratio was studied from 1:1 to 1:15 (Figure 3), maintaining the other experimental conditions (room temperature and PW_12_ amount). As expected, the increase in acetone amount instigated a higher conversion of glycerol, as well as the solketal production. Using 1:1, no conversion of glycerol was observed. Using 1:12 and 1:15 molar ratios, the glycerol acetalization reached the highest glycerol conversion after only 5 min of reaction, i.e., 95.4% of conversion using a ratio of 1:12, and even a near complete conversion (99.2%) using 1:15, with a solketal selectivity of 89.2% and 97.0%, respectively. Therefore, the study of the catalyst amount (varying from 1% to 3% based on the glycerol weight) was performed using the highest amount of acetone and at 25 °C (Figure 4). The decrease of catalyst amount from 3% to 1% (based on glycerol weight) was demonstrated to proportionally have an effect on solketal selectivity, which decreased from 97.0% to 89.8% and also in the glycerol conversion (from 99.2% to 90.2%, respectively). Previously, Da Silva et al. obtained 63% of the conversion and 97% of the selectivity to solketal after 2 h of reaction using similar reaction conditions (1:15 glycerol/acetone ratio, at room temperature). This lower conversion was achieved due to the lower amount of active center used, i.e., 1.0 mol % of PW_12_ [16]. 

It is well-established that the selectivity of acetalization reactions depend on the strength of the acidic centers [3,20]; therefore, it would be interesting to explore a series of heteropolyacids (HPAs) as Brønsted acid catalysts. Figure 5 presents the catalytic data obtained using the various HPAs (using an amount of 3% of each based on glycerol weight) for glycerol acetalization with acetone (glycerol: acetone ratio 1:15) under room temperature, obtained after 5 min of reaction. The activity of the various HPAs followed the order: PW_12_ (99.2%) > PMo_12_ (91.4%) > SiW_12_ (90.7%). This similar catalytic behavior obtained for the phosphomolybic and silicotungstic HPAs can be related with a similar acid strength, from which the PW_12_ stands out. [21] All catalysts used exhibited a high selectivity (from 85.7% to 97.0%) towards the solketal product. The high effectivity of Brønsted acid catalysts for glycerol acetalization can be explained by the reaction mechanism (Figure 1). In a first step, occurs the formation of a glycerol–acetone adduct, which turns a hemiketal interacting with the Brønsted acid sites and generating a carbonium ion upon dehydration. This carbocation is stabilized by a nucleophilic attack of the secondary or terminal hydroxyl group of glycerol producing the five-membered (solketal) and six-membered (acetal) products. The short lifetime of the carbocation compared to the hemiketal favored the formation of the solketal [18,20,21].

Table 1 presents the various reported results using the polyoxometalate-based catalysts for the reaction of glycerol acetalization using acetone under solvent-free systems. In all the reported examples, the selectivity for the solketal was higher than 90% and most of these studies used the Keggin-type polyoxotungstate [PW_12_O_40_]^3−^

Higher conversions of glycerol were obtained using the cesium salt of the phosphotungstate compared to the phosphotungstic acid [2,14,22]. However, this last presents a slightly lower selectivity for the desirable product. Furthermore, the catalytic activity of the cesium compound was maintained after its immobilized in the mesoporous silica KIT-6. [2] On the other hand, the activity of the phosphotungstic acid was increased after its immobilization, also in a silica support material. [21] The catalyst ⎨H_20_⎬-355 treated at 355 °C presents probably a structural modification to the initial prepared polyoxometalate; however, its activity and selectivity was similar to the Keggin-type phosphotungstates. Compared to the previous published works using polyoxometalates as catalysts for the acetalization of glycerol without the need of a solvent addition, the heteropolyacids used, H_3_PW_12_O_40_ and H_3_PMo_12_O_40_, demonstrated to be more efficient, since the near complete glycerol conversion was achieved after only 5 min, instead of the 2 h needed by Da silva et al. [16]. The higher efficiency obtained in this work was due to the application of the optimized conditions conciliating the proper ratio of glycerol, acetone, catalyst and the use of room temperature (25 °C).

**Table 1 molecules-27-06573-t001:** Polyoxometalates used as catalysts for glycerol acetalization in the absence of an auxiliary solvent.

Catalyst	Ratio of Glycerol/Acetone	Temp. (°C)	Time (h)	Conversion (%)	Selectivity to Solketal (%)	Ref.
**Cs_2.5_H_0.5_PW_12_O_40_**	1:6	RT	1	94	>90	[2]
**Cs_2.5_H_0.5_PW_12_O_40_@KIT-6**	1:6	RT	0.25	95	>90	[2]
**H_3_PW_12_O_40_**	1:10	RT	2	58	97	[16]
**H_3_PW_12_O_40_**	1:10	60	1	47.8	97.5	[22]
**H_3_PW_12_O_40_@SiO_2_**	1:6	70	4	97	97	[21]
**⎨H_20_⎬-355**	1:2	RT	0.67	98	>98	[23]
**H_3_PW_12_O_40_**	1:15	RT	0.08	99.2	97	This work
**H_3_PMo_12_O_40_**	1:15	RT	0.08	91.4	94	This work
**H_4_SiW_12_O_40_**	1:15	RT	0.08	90.7	85.7	This work

RT means room temperature; ⎨H_20_⎬-355 abbreviated from [P_8_W_60_Ta_12_(H_2_O)_4_(OH)_8_O_236_]·128H_2_O calcinated at 355 °C.

A higher number of heterogeneous catalysts (without polyoxometalate structuress as active centers) can be found in the literature for the glycerol acetalization reaction without the use of solvent. Table 2 presents the various catalytic materials previously used. These are mainly porous MOFs, zeolites, silica materials and resins. One of the best results was obtained by the MOF structure UiO-66(Hf) at room temperature, achieving 95% of conversion of glycerol and 97% of selectivity to solketal after only 1 h [20]. Still at room temperature, the nano oxide zirconium SO_4_^2−^/ZrO_2_ and silica TiO_2_@SiO_2_ materials achieved similar results after 1.5 and 3 h, respectively [24,25]. The Amberlyst-15 also demonstrated to be an active catalyst for this reaction, achieving 95% of conversion and 99% of selectivity for solketal after only 15 min; however, in this case, a 70 °C was needed to be used instead of room temperature [26].

Conciliating the reported results of the heterogeneous catalysts present in Table 2 with the results obtained in this work with various heteropolyanions, the immobilization of these last into porous MOFs and porous silica materials are planned to be performed in the near future to turn the high efficiency of H_3_PW_12_O_40_ and H_3_PMo_12_O_40_ here obtained into a recovered, separable and recycled catalyst.

## 3. Materials and Methods

### 3.1. Materials

All chemicals were purchased from commercial sources and used without further purification. Glycerol (HOCH_2_CH(OH)CH_2_OH, 99.92%, Fluka), acetone ((CH_3_)_2_CO, ≥99%, Sigma-Aldrich), phosphotungstic acid hydrate (H_3_[PW_12_O_40_]·*n*H_2_O, for microscopy, Fluka), phosphomolybdic acid hydrate (H_3_[PMo_12_O_40_]·*n*H_2_O, for microscopy, Sigma-Aldrich), silicotungstic acid (H_4_[SiW_12_O_40_]·*n*H_2_O ≥99.9%, Aldrich), toluene anhydrous (C_6_H_5_CH_3_, 99.8%, Sigma-Aldrich) and ethanol (CH_3_CH_2_OH, ≥99.8%, Fluka).

### 3.2. Catalytic Experiments

A typical acetalization reaction of glycerol with acetone was carried out under the air in a closed borosilicate 5 mL vessel, equipped with a magnetic stirrer and immersed in a thermostatically controlled liquid paraffin bath (25–60 °C). For each run, the solution based on glycerol and acetone was preheated to the chosen temperature (25–60 °C) during 10 min and then the catalyst (1–3 wt%, based on the glycerol weight) was added, starting the reaction. The reaction evolution and products’ analysis were controlled by GC-FID Bruker 430-CC and a SUPRAWAX-280 capillary column (30 m length × 0.25 mm i.d. × 0.25 µm film thickness) and hydrogen as gas carrier with a 30 mL.min^−1^ flow rate. The GC thermal condition started with 2 min at 80 °C, ramped with 20 °C/min rate to 260 °C with a hold time of 2 min, and then finalized with 30 °C/min rate to 290 °C with a hold time corresponding to 3 min. The injector and detector temperatures were held at 260 °C and 290 °C, respectively. Toluene was used as internal standard. The glycerol conversion was determined by the relation of the peak area of the glycerol and internal standard: conversion (%) = (A_solketal_/(A_glycerol_ + A_Solketal_)) × 100. Selectivity for solketal = (A_solketal_/(A_Acetal_ + A_Solketal_)) × 100, acetal was the secondary product obtained (Figure 1).

## 4. Conclusions

The studies using different Keggin-type HPAs as homogeneous catalysts for glycerol acetalization with acetone without the need of an additional solvent demonstrated a highly conversion and selectivity to produce solketal. The activity of the various HPAs followed the order: PW_12_ (99.2%) > PMo_12_ (91.4%) > SiW_12_ (90.7%) as a result of the strong acidic sites. Only 5 min of reaction were needed to achieve 97% of solketal product in the presence of the PW_12_ as a catalyst at room temperature. The acetalization reaction revealed to depend on the temperature, obtaining a higher glycerol conversion into solketal at room temperature than at 60 °C. Moreover, increasing the acetone amount instigated a higher conversion of glycerol, as well as a solketal production as the main product. The catalyst amount had also an effect on solketal selectivity and glycerol conversion, promoting a decreasing of conversion and solketal production when the amount of catalyst decreased from 3% to 1% (based on glycerol weight). A fast glycerol conversion and high selectivity to solketal in sustainable conditions was observed, under solvent-free conditions and at room temperature. Overall, the studies here reported open the possibility of glycerol acetalization using acetone under a more sustainable catalytic system that promote a high efficiency to HPAs at very short reaction time.

## Data Availability

Not available.

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
