# Peer review of "Easy and Fast Production of Solketal from Glycerol Acetalization via Heteropolyacids"

_molecules, 2022, doi:10.3390/molecules27196573_

Round 1

Reviewer 1 Report

Dear authors,

After read the manuscript, I would like to recommend to submit it as short communication to another journal such as "Applied Sciences". I believe the results are interesting, but still preliminar and also, the manuscript has a lack of discussion. More tables comparing the results with those available in the literature would increase significantly the quality of the paper.

Author Response

Authors Response: The authors acknowledge the reviewer for this point that it helps to improve the quality of our manuscript. Tables 1 and 2 conciliating reported results in the area of glycerol acetalization, using polyoxometalates (table 1) and heterogeneous catalysts (Table 2) were prepared and introduced in the manuscript in pages 7 to 9, highlighted at orange. Also, a comparative study of catalysts efficiency and reaction conditions were performed and discussed.

Reviewer 2 Report

The manuscript has some novelty, but in my opinion , the results could be more discussed , using several data available in the literature. The discussion is weak point and the conclusions are predictable. I recommend you increase the number of references and add some tables, transforming your paper in a short communication or to include a kinetical study to increase the quality of the paper

Author Response

Authors Response: The authors acknowledge the reviewer for this point that it helps to improve the quality of our manuscript. Tables 1 and 2 conciliating reported results in the area of glycerol acetalization, using polyoxometalates (table 1) and various heterogeneous catalysts (Table 2) were prepared, discussed and introduced in the manuscript in pages 7 to 9. Further, comparative study of catalysts efficiency and reaction conditions between published results and obtained results were performed and discussed in the manuscript.

Reviewer 3 Report

 Julião et al. studied the production of solketal from glycerol acetalization over heteropolyacids catalysts. The reactional parameters such as the reactants stoichiometry, catalyst concentration and reaction temperature were investigated. However, the paper might be published after major revisions noted below.

1. Introduction section does not show the comprehensive research progress of the the applications of glycerol and heteropolyacids. The background knowledge on this area must be further reviewed, by consulting the recently published articles, for example: Journal of the Taiwan Institute of Chemical Engineers, 2022, 133: 104277; Korean Journal of Chemical Engineering, 2020, 37:955-960; Journal of the Taiwan Institute of Chemical Engineers Volume 128, November 2021, Pages 388-399; Chemical Engineering Journal Volume 316, 15 May 2017, Pages 797-806; Applied Catalysis B: Environmental Volume 309, 15 July 2022, 121247; Molecular Catalysis Volume 516, November 2021, 111975.

2. 2.2. Catalytic experiments The formula of glycerol conversion and product selectivity should be added.

3. The authors studied the effects of reactional parameters on the performances. Actually, the tungstophosphoric acid (PW12), silicotungstic acid (SiW12) and phosphomolybdic acid (PMo12) show different acidity, and the acidity has an important impact on the performance. Therefore, the author should study the acidity of the catalyts and establish the relationship between the performance and acidity.

4. Actually, the catalysts in this paper is very normal, the nolvelty of this papaer is limited. Did the author study author catalysts? For example, the heteropolyacids supported catalysts.

5. The authors should compared the previous reported results about other catalysts with this paper, i suggest listing a table for comparison.

From the above, the authors should perform a major revision of the manuscript taking into consideration at least the comments above. Other small issues along the manuscript resulting from a less correct use of the English language should also be considered.

Author Response

  1. Introduction section does not show the comprehensive research progress of the the applications of glycerol and heteropolyacids. The background knowledge on this area must be further reviewed, by consulting the recently published articles, for example: Journal of the Taiwan Institute of Chemical Engineers, 2022, 133: 104277; Korean Journal of Chemical Engineering, 2020, 37:955-960; Journal of the Taiwan Institute of Chemical Engineers Volume 128, November 2021, Pages 388-399; Chemical Engineering Journal Volume 316, 15 May 2017, Pages 797-806; Applied Catalysis B: Environmental Volume 309, 15 July 2022, 121247; Molecular Catalysis Volume 516, November 2021, 111975.

Authors Response: The authors acknowledge the reviewer for this point that it helps to improve the quality of our manuscript. The introduction of the manuscript was improved by presenting the reported works using polyoxometalates as catalysts for the glycerol acetalization, introducing novel references in the manuscript. Further, the main goals of this work was also described here. All the alterations are highlighted at green in the introduction section.

  1. 2.2. Catalytic experiments The formula of glycerol conversion and product selectivity should be added.

Authors Response: The authors acknowledge the reviewer for these comments. The formula for glycerol conversion and product selectivity were introduced in the manuscript in experimental part, section 2.2, highlighted at green.

  1. The authors studied the effects of reactional parameters on the performances. Actually, the tungstophosphoric acid (PW12), silicotungstic acid (SiW12) and phosphomolybdic acid (PMo12) show different acidity, and the acidity has an important impact on the performance. Therefore, the author should study the acidity of the catalyts and establish the relationship between the performance and acidity.

Authors Response: The authors acknowledge the reviewer; however, the study comparing catalytic activity and acidity was performed in this work and presented in section 3 page 6: “The activity of the various HPAs followed the order: PW12 (99.2%) > PMo12 (91.4%) > SiW12 (90.7%). This similar catalytic behavior obtained for the phosphomolybdate and silicotungstate HPAs can be related with a similar acid strength, from which the PW12 stands out.[19]”

  1. Actually, the catalysts in this paper is very normal, the nolvelty of this papaer is limited. Did the author study author catalysts? For example, the heteropolyacids supported catalysts.

Authors Response: The authors acknowledge the reviewer for these comments. The novelty of this work was highlighted in the introduction section. The novel contribution of this work for the scientific area of glycerol  acetalization was described. Tables 1 and 2 conciliating reported results in the area of glycerol acetalization with acetone, using polyoxometalates and the only work reporting supported heteropolyoxometalate (Table 1) and various heterogeneous catalysts (Table 2) were prepared, discussed and introduced in the manuscript in pages 7 to 9. Further, comparative study of catalysts efficiency and reaction conditions between published results and obtained results in this work were performed and discussed in the manuscript.

  1. The authors should compared the previous reported results about other catalysts with this paper, i suggest listing a table for comparison.

Authors Response: As answered in your previous question, reported results were presented, discussed and compared with the results obtained in this work. Tables 1 and 2 were prepared as introduced in the manuscript

Round 2

Reviewer 1 Report

It can be published as short paper

Reviewer 2 Report

The manuscript can be accepted as short communication

Reviewer 3 Report

The paper can be accepted.